# The Negative Impact of Night Shifts on Diet in Emergency Healthcare Workers

**DOI:** 10.3390/nu14040829

**Published:** 2022-02-16

**Authors:** Jean-Baptiste Bouillon-Minois, David Thivel, Carolyne Croizier, Éric Ajebo, Sébastien Cambier, Gil Boudet, Oluwaseun John Adeyemi, Ukadike Chris Ugbolue, Reza Bagheri, Guillaume T. Vallet, Jeannot Schmidt, Marion Trousselard, Frédéric Dutheil

**Affiliations:** 1Emergency Department, CHU Clermont-Ferrand, Université Clermont Auvergne, CNRS, LaPSCo, Physiological and Psychosocial Stress, F-63000 Clermont-Ferrand, France; eric.ajebo_diko@etu.uca.fr (É.A.); jschmidt@chu-clermontferrand.fr (J.S.); 2Research Center in Human Nutrition, Laboratory AME2P, Université Clermont Auvergne, F-63120 Aubière, France; david.thivel@uca.fr; 3Hematology Department, CHU Clermont-Ferrand, Université Clermont Auvergne, F-63000 Clermont-Ferrand, France; ccroizier@chu-clermontferrand.fr; 4Clinical Research and Innovation Direction, CHU Clermont-Ferrand, F-63000 Clermont-Ferrand, France; scambier@chu-clermontferrand.fr; 5Department of Psychology, Université Clermont Auvergne, CNRS UMR 6024, LaPSCo, F-63000 Clermont-Ferrand, France; gil.boudet@uca.fr (G.B.); guillaume.vallet@uca.fr (G.T.V.); 6Ronald O. Perelman Department of Emergency Medicine, NYU School of Medicine, New York University Langone Health, New York, NY 10012, USA; oluwaseun.adeyemi@nyulangone.org; 7School of Health and Life Sciences, Institute for Clinical Exercise & Health Science, University of the West of Scotland, Glasglow G72 0LH, UK; u.ugbolue@uws.ac.uk; 8Department of Exercise Physiology, University of Isfahan, Isfahan 81746-73441, Iran; will.fivb@yahoo.com; 9French Armed Forces Biomedical Research Institute, IRBA, Neurophysiology of Stress, F-91223 Brétigny-sur-Orge, France; marion.trousselard@gmail.com; 10Occupational and Environmental Medicine, CHU Clermont-Ferrand, Université Clermont Auvergne, CNRS, LaPSCo, Physiological and Psychosocial Stress, WittyFit, F-63000 Clermont-Ferrand, France; fdutheil@chu-clermontferrand.fr

**Keywords:** nutrients, work, well-being, quality of life, prevention, public health

## Abstract

Despite the consequences of night-shift work, the diet of night-shift workers has not been widely studied. To date, there are no studies related to food intake among emergency healthcare workers (HCWs). We performed a prospective observational study to assess the influence of night work on the diet of emergency HCWs. We monitored 24-h food intake during a day shift and the consecutive night, and during night work and the daytime beforehand. We analyzed 184 emergency HCWs’ food intakes. Emergency HCWs had 14.7% lower (−206 kcal) of their 24-h energy intake during night shifts compared to their day-shift colleagues (1606.7 ± 748.2 vs. 1400.4 ± 708.3 kcal, *p* = 0.049) and a 16.7% decrease in water consumption (1451.4 ± 496.8 vs. 1208.3 ± 513.9 mL/day, *p* = 0.010). Compared to day shifts, night-shift had 8.7% lower carbohydrates, 17.6% proteins, and 18.7% lipids. During the night shift the proportion of emergency HCWs who did not drink for 4 h, 8 h and 12 h increased by 20.5%, 17.5%, and 9.1%, respectively. For those who did not eat for 4 h, 8 h and 12 h increased by 46.8%, 27.7%, and 17.7%, respectively. A night shift has a huge negative impact on both the amount and quality of nutrients consumed by emergency healthcare workers.

## 1. Introduction

According to the International Labor Organization, night work is defined as “all work which is performed during a period of not less than seven consecutive hours, including the interval from midnight to 5 a.m.” [1]. In 2020 in Europe, 22% of men and 11% of women worked on shifts that included night work [2], including healthcare workers (HCWs) [2,3]. Night work has many health consequences, from disturbance of the circadian rhythm [4], to obesity and cardiometabolic disorders [5,6,7,8,9,10]. Interestingly, eating disorders are also associated with those pathologies, potentially enhancing the synergistic effect between night work and eating disorders [11]. Additionally, night-shift workers tend to have more irregular eating habits than their day colleagues [12]. Night work also induces a conflict between socially determined diurnal mealtimes, eating habits, the biological rhythm of hunger, satiety, and metabolism [13,14]. Some findings showed that meal timing and meal size have an impact on cognitive performance and subjective sleepiness among night-shift workers. Some programs proposed to avoid large meals during the beginning of a night shift and to opt for a small snack to improve performance during the night [15]. Lastly, eating behavior can be influenced by sociodemographic criteria such as age and gender [16], and by occupational characteristics such as experience [16] and workload [17,18]. Although nurses have been widely studied, no conclusion can be made and the existing epidemiological evidence on the relationship between night-shift work of nurses and their dietary habits is inadequate to draw any definite conclusions [19]. Emergency HCWs are a perfect example for studying the impact of night shifts [20]. Indeed, emergency departments (ED) are open 24 h a day, 365 days a year. Furthermore, emergency HCWs are working under stressful conditions such as overcrowding—lack of beds in hospitals [20], life-threatening emergencies [21], the wait for possible disasters [22] with consequences on biomarkers of stress [23,24]. Furthermore, the availability of food is much less important during the night compared to the day in a hospital. It is rarely fresh but often packaged reheated food with a bad presentation. However, food preparation and presentation appears to influence student consumption of school food and adult perception of school meal quality [25]. Given that emergency HCWs work under the time pressure of urgent care, they must both take care of patients and themselves i.e., finding the time to eat and drink. To the best of our knowledge, there are no studies that assessed the influence of night work on the diet among the population of emergency HCWs, nor in relation to their occupational characteristics.

Therefore, the main objective of our study was to assess the influence of night work on the diet of emergency HCWs. Secondary objectives were to study the impact of socio-demographic, and occupational characteristics on the diet of emergency HCWs.

## 2. Materials and Methods

### 2.1. Study Design

We performed a prospective nationwide observational study in five French hospitals—two university hospitals and three non-university hospitals. The main inclusion criterion was to work as an emergency HCW. Exclusion criteria were refusal to participate and pregnancy. This study is part of the SEEK protocol [20,22]. We obtained the ethical approval from Ethics Committee South-East I (DC-2014-2151) and the protocol was registered on ClinicalTrials.gov as number NCT02401607. Each participant had their food intake monitored twice over 24 consecutive hours: (1) during a day shift (from 8:30 a.m. to 6:30 p.m.) + the night (no work, 6:30 p.m. to 8.30 a.m.), and (2) during a rest day (from 8:30 a.m. to 6:30 p.m.) and a night shift (6:30 p.m. to 8:30 a.m.) (Figure 1). Participants also had to complete a questionnaire on their sociodemographic characteristics.

### 2.2. Outcomes

Participants were asked to complete a 24-h dietary recall that was explained to them by a member of the investigation team. The participants were asked to indicate as precisely as possible all the details regarding the food ingested at each meal and in-between meals. The diaries were reviewed afterward with the participants during a dedicated interview. We next used Nutrilog^®^ (version 3.2, Nutrilog, Marans, France), a diet and nutrition software for health care professionals, to translate participants’ answers on nutrients intake using nutrition table Ciqual^®^ on this software [26]. We retrieved total energy intake (kcal), glucids (grammes and % energy intake, and sugar), lipids (grammes and % energy intake, and simple, mono- and poly-unsaturated fatty acids), protids (grammes and % energy intake), calcium, cholesterol, fibres, and water. We further looked at time of intake and more specifically we calculated number of emergency HCWs that did not eat or drink for more than 4, 8, and 12 consecutive hours.

Participants had also to complete a short questionnaire to retrieve their sociodemographic (age, gender, marital status) and work characteristics (occupation, job seniority, and setting—university hospital or not).

### 2.3. Statistics

The main judgment criterion was food intake as quantitative variables. From personal experience and observation in a preliminary non-published study among 10 emergency HCWs, there is at least a 20 ± 20% decrease in food intake during a night shift compared to a day shift, because of the increased working demands during a night shift due to fewer staff. Using this difference as the main outcome, we calculated that a sample size of 10 participants allowed a statistical power greater than 80% with an alpha level less than 5%.

Statistical procedures were performed with Stata software (v17, College Station, TX, USA). Quantitative variables were expressed as mean ± standard deviation (SD), and qualitative variables were expressed as number (%). The Gaussian distribution for each food intake variable was assessed by a Shapiro–Wilk test. Mean levels of each 24 h food intake variable were compared between night and day shifts using Wilcoxon (signed rank) tests. Association between food intake, sociodemographic (age, gender, marital status) and work characteristics (occupation, job seniority, and setting—university hospital or not), were assessed using Chi2 for categorical variables, and Spearman correlations for quantitative variables. The influence of shift condition, sociodemographic and work characteristics with each food intake variable were further assessed using univariate regressions—linear regressions for quantitative variables (coefficient and 95 confidence intervals—95% CI) and logistic regression for qualitative variables (odds ratio and 95% CI). Significance was set at the *p* < 0.05 level.

## 3. Results

We included 192 emergency HCWs from five hospitals. Four emergency HCWs were excluded because pregnancy and four for incomplete data. In total, we analyzed 184 emergency HCWs (Figure 1). Table 1 presents sociodemographic characteristics of the participants.

### 3.1. Main Objective: Impact of Night Shift on Alimentation

#### 3.1.1. Quantitative Data

The night shift group of emergency HCWs reported a 14.7% (−206 kcal) lower 24-h energy intake compared to the day shift group (1606.7 ± 748.2 vs. 1400.4 ± 708.3 kcal, *p* = 0.049). Compared to day shifts, there were a 8.7% decrease among the night shift group of carbohydrates (178.8 ± 81.5 vs. 163.3 ± 86.0 g/day, *p* = 0.120), 17.6% of proteins (80.6 ± 30.3 vs. 66.4 ± 36.4 g/day, *p* < 0.001), 18.7% of lipids (75.1 ± 35.9 vs. 61.13 ± 32.7 g/day, *p* = 0.030), 18.2% of simple fatty acids (31.8 ± 16.1 vs. 26.0 ± 13.4 g/day, *p* = 0.056), 19.1% of monounsaturated fatty acids (25.6 ± 14.2 vs. 20.7 ± 14.2 g/day, *p* = 0.049), 20.9% of polyunsaturated fatty acids (7.7 ± 4.8 vs. 6.13 ± 4.5 g/day, *p* = 0.050), 13.9% of calcium (665.6 ± 249.3 vs. 573.3 ± 297.9 mg/day, *p* = 0.049), 12.5% of cholesterol (318.7 ± 238.9 vs. 278.7 ± 243.0 mg/day, *p* = 0.161), 11.7% of fiber (15.8 ± 7.1 vs. 13.9 ± 8.2 g/day, *p* = 0.171) and 16.7% of water consumption (1451.4 ± 496.8 vs. 1208.3 ± 513.9 mL/day, *p* = 0.010). (Table 2 and Figure 2).

#### 3.1.2. Qualitative Data

Overall during shifts, there were 38% of emergency HCWs who did not drink for 4 h, 19% for 8 h, and 5% for 12 h; and 46% who did not eat for 4 h, 27% for 8 h, and 9% for 12 h. More specifically, the night shift group of emergency HCWs reports a proportion of participants who did not drink for 4 h, 8 h and 12 h increased by 20.5% (50.0% vs. 29.5%, *p* = 0.015), 17.5% (29.3% vs. 11.5%, *p* = 0.009), and 9.1% (10.3% vs. 1.3%, *p* = 0.018) respectively. Similarly, the proportion of emergency HCWs who did not eat for 4 h, 8 h and 12 h was higher in the night shift group by 46.8% (72.4% vs. 25.6%, *p* < 0.001), 27.7% (43.1% vs. 15.4%, *p* < 0.001), and 17.7% (19.0% vs. 1.3%, *p* < 0.001) (Table 2 and Figure 3) respectively.

### 3.2. Regression Analysis

#### 3.2.1. Quantitative Variables

Energy intake is significantly lower among females compared to males (coefficient = 340.1, 95% CI 107.5 to 572.7, *p* = 0.004) and among non-university hospitals compared to university hospitals (378.5, 143.9 to 613.1, *p* = 0.002). Consumption of lipids was decreased by night work (−3.34, −6.59 to −0.09, *p* = 0.04), and increased by age (0.19, 0.04 to 0.35, *p* = 0.02), and in workers who were single (3.80, 0.36 to 7.24, *p* = 0.03). Simple fatty acid consumption was decreased by night work (−5.22, −10.34 to 0.1, *p* = 0.051), and increased with age (0.27, 0.02 to 0.52, *p* = 0.023). Cholesterol consumption increased with age (4.78, 1.21 to 8.34, *p* = 0.009), male (75.8, 1.71 to 149.8, *p* = 0.04), and seniority (5.03, 0.27 to 9.80, *p* = 0.038). Night work did not statistically influence other nutrition variables (Figure 4 and Appendix A).

#### 3.2.2. Qualitative Variables

Night shifts triggered a rise in the risk of not drinking for more than 4 h, 8 h and 12 h by 2.4 (OR = 2.39, 95% CI 1.11 to 5.16, *p* = 0.015), 3.2 (3.18, 1.20 to 8.82, *p* = 0.009) and 8.9 (8.88, 1.04 to 413.8, *p* = 0.018) respectively compared to day shifts. Furthermore, night shifts increased the risk of not eating for more than 4 h, 8 h and 12 h by 7.6 (7.61, 3.31 to 17.7, *p* < 0.001), 4.2 (4.17, 1.74 to 10.22, *p* < 0.001) and 18 (18.0, 2.43 to 785.3, *p* < 0.001), respectively, (Figure 3 and Figure 4, and Appendix A). Except for night shift, the regression analysis did not find any impact on sociodemographic, location of work or seniority.

## 4. Discussion

We demonstrated that nightshift has a negative impact on both the amount and quality of nutrients intake among emergency HCWs. Furthermore, 27% of emergency HCWs do not have time to eat during more than 8 consecutive hours of the night shifts, and they do not even have time to drink for nearly one fifth of them. Main other influencing variables were seniority and location of centers.

### 4.1. The Negative Impact of Night Shift on Food Intake

Emergency HCWs working during a night shift have a slighter 24-h energy intake compared to those working during a day shift, but also compared to daily recommended diet [27]. Literature is not consistent about these results. Although we did not find any study about the impact of night shifts among emergency physicians, there is some among nurses with controversial results. Some found a lower energy intake among night workers versus day workers [28,29] while others did not find any significant differences [30,31,32]; however, another study found higher food intake among night workers versus day workers [33]. Although we do not study the energy intake the day after the shift, it seems that this low food intake is compensated for on the day of recovery after work [29]. We can explain this low food intake in our study. First, emergency HCWs are subjected to daily stress due to overcrowding of EDs [34], lack of beds available in the hospital, and life-threatening emergencies [35,36]. However, the impact of stress on eating is controversial. Indeed, while acute stress is able to induce dietary restriction, chronic stress will induce food intake signal thus providing a better response to future stressful events [37]. Many pathophysiological pathways have been studied, especially the impact of ghrelin and leptin as biomarkers of acute stress [38,39,40]. It would be interesting to study those biomarkers among emergency HCWs. Secondly, sleep restriction and disturbed nychthemeral rhythm promote increased energy intake [19,41].

### 4.2. The Amount and Quality of Nutrient Intake and Shifts

The quality and choice of food during the night shift differs from the day shift [19]. This could induce restrictive behavior. Our results are in accordance with this. Indeed, emergency HCWs eat fewer carbohydrates and lipids during night shifts than their daytime colleagues [28,42]. Those results are also in agreement with the literature. Furthermore, it seems that the main carbohydrate intake during night shifts include the excessive consumption of sweets and sugary drinks [43,44,45]. Concerning lipids, night workers, and especially permanent night-shift workers, are at risk of dyslipidemia [9]. However, in our study, emergency HCWs eat less lipid and fat during night shifts than during day shifts. This is in line with literature [28,29,43]. Another possible explanation is the lack of time to eat during both day and night shifts. It seems that this lack of time leads to an increase in consumption of cold foods and fast foods [31,44,45], which predisposes individuals to obesity and cardiovascular disorders [46,47]. We also showed a low protein consumption during night shifts—type of position held, overwork and the quality of sleep are risk factors for low protein consumption [27,47]. Our results showed a low consumption of water consumed by emergency HCWs. Some of them do not even consume fluids during the entire night shift, i.e., during more than 12 h [28,48]. It seems that coffee/tea and sugary drinks are the preferred drinks mostly consumed during night work [28,33,49].

### 4.3. Other Influencing Variables

We showed a link between age, emergency HCWs’ position held and the consumption of lipids, fat, and cholesterol. The more experience they have, the more lipids, fat, and cholesterol they consume. This is in line with a study that shows that eating behavior is influenced by seniority, and age [50]. It seems that experienced workers, with staggered schedules, eat more during the night, but mainly snacks rich in sugar, fat, and energy [50,51]. We also found that males have a higher energy, cholesterol, and protein intake, but also are less able to spend more than 4 h without eating. This finding is not in agreement with the literature. Indeed, night work induces an increase in the consumption of sugary and fatty foods, possibly in response to stress among emotional eaters, who are most often women [52,53]. University hospitals are larger than non-university hospitals, so we could imagine that emergency HCWs have less time to eat because ED is more overcrowded. However, we found that working in a large health center would have a positive influence on the diet of emergency HCWs. According to the literature, night workers in large hospital centers tend to consume more “junk food”, which is easily accessible and more convenient to eat [28]. Our study is the first to examine the influence of night work on the diet of emergency physicians. Indeed, the subject often explored nurses or even more often industries in Asia with a population having a very high energy expenditure and a very low socio-economic level. Most of these studies have a low level of evidence (non-representative samples, unsystematic literature reviews) and are difficult to transpose in our population due to very different eating habits and high standards of living [19]. Another potential variable could be the sleep that we did not study the impact of. Indeed, previous literature suggests that night-shift workers obtain less sleep compared to day-shift workers. Thus, if night-shift workers have a longer duration awake, they will be hungrier, and so more intake [54].

### 4.4. Limitations

Our study has some limitations. Firstly, the analyses of the dietary data were not necessarily done in the same subjects for day and night as we had more day shifts and fewer night shifts. Secondly, we worked on questionnaires that utilized self-reported information that may have contained some imprecision in the description of the food portions consumed as well as over- or under-declarations on the part of the participants completing the questionnaires. Although our data were collected via a self-assessment questionnaire with self-reported responses, the general methodological quality of our study was good and the risk of bias across participants was reduced through the use of a validated scale presented to every emergency HCW before the beginning of the data collection session. Another limitation is that we did not study the impact of cognition. Indeed, the impact on memory in case of a decrease of attentiveness or responsiveness could impact negatively the ability to fill in the questionnaire. By and large, while quality assurance protocols were put in place for the data collection process, it was difficult to micro-manage the different centers. Lastly, strict inclusion criteria were applied in order to extract data from a targeted population of emergency HCWs and thus answer our research questions. Continuous monitoring of food intake may be relevant and mandatory in future research. On the other hand, few studies consider the diet of emergency HCWs, and when this is the case, it concerns nurses [55]. Our study is therefore the first to look at the nutrition of emergency physicians during night shifts. Unfortunately, we did not study the food intake the day following the night shift. We previously showed that a night shift has a prolonged effect on some outcomes such as biomarkers of stress [23,24]. Further studies assessing the food intake during rest after a night shift could be relevant. Finally, very few studies have looked at the impact of night work on emergency physicians [24]. It would, therefore, be interesting to conduct more prospective and interventional studies to try to explore all the underlying mechanisms and subsequently put in place health-promotion and/or preventive strategies.

## 5. Conclusions

We showed that emergency healthcare workers working during nightshift have a lower amount and quality of nutrient intake. They seem to eat less and less healthily. Furthermore, sometimes some emergency HCWs go for long hours without eating. One fifth of them do not even drink for more than 8 consecutive hours of night shifts. Further studies assessing the food intake during rest after night shifts could be relevant. Preventive policies should target those workers.

## Figures and Tables

**Figure 1 nutrients-14-00829-f001:**
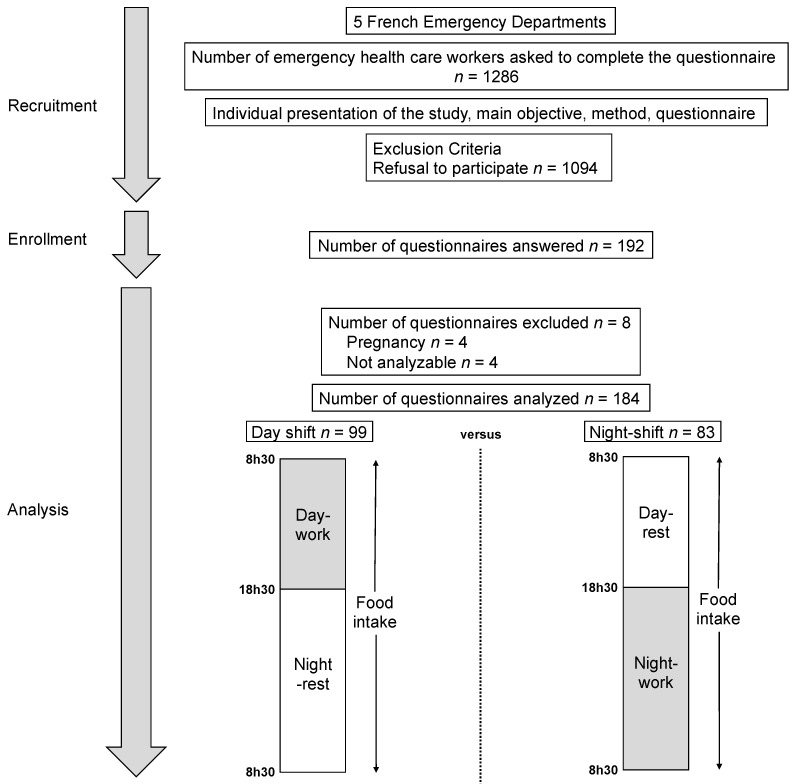
Study design. Among five French emergency departments we were able to recruit 192 emergency health care workers. Eight were excluded because of pregnancy or no data completion. Food intake was studied for 99 emergency health care workers during a day shift and 83 during a night shift.

**Figure 2 nutrients-14-00829-f002:**
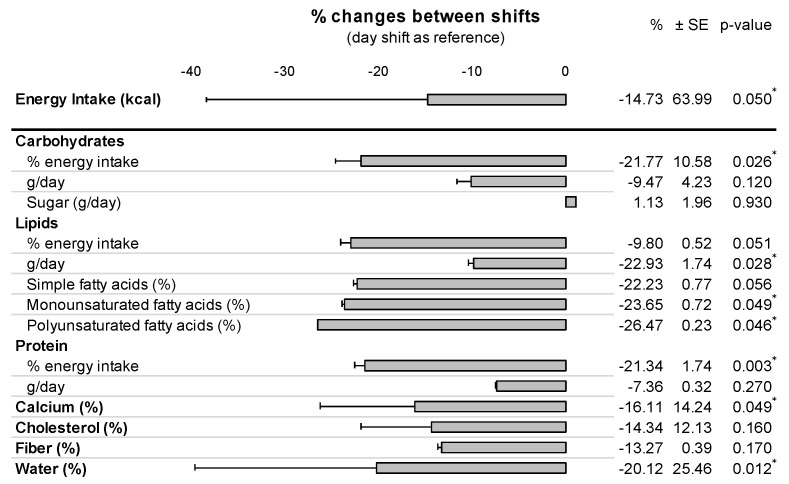
Food intake variations between night and day shifts. Differences in total energy intake, carbohydrates, lipids, protein, calcium, cholesterol, fiber, and water consumptions between day and night shifts among 184 questionnaires. Differences expressed in percentage with day shift as reference. Results: a night shift induces a decrease of all components related to daily ingesta except on sugar. SE standard error; *p*-value with * are significant (if <0.05).

**Figure 3 nutrients-14-00829-f003:**
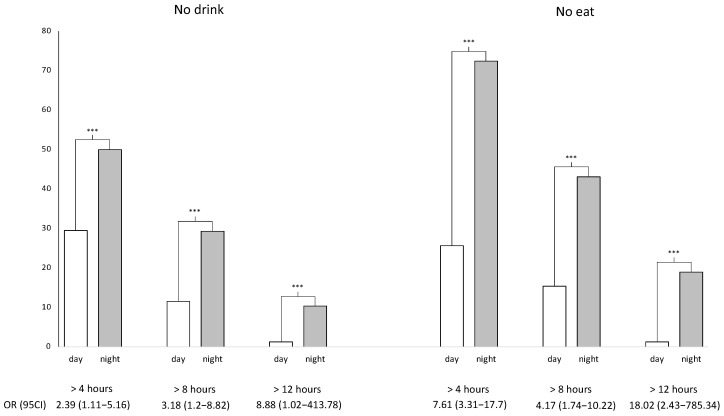
Risk of no-intakes for emergency health care workers during shifts. Percentage of participants declaring not drinking/not eating, over 4-, 8-, or 12-h during night- and day-shifts. A total of 136 questionnaires were analyzed. Odds ratio (OR) was used to access the impact of nightshifts using day-shifts as reference. 95% CI = 95% confidence intervals; *** represent *p*-value which are all <0.018 (significant if <0.05).

**Figure 4 nutrients-14-00829-f004:**
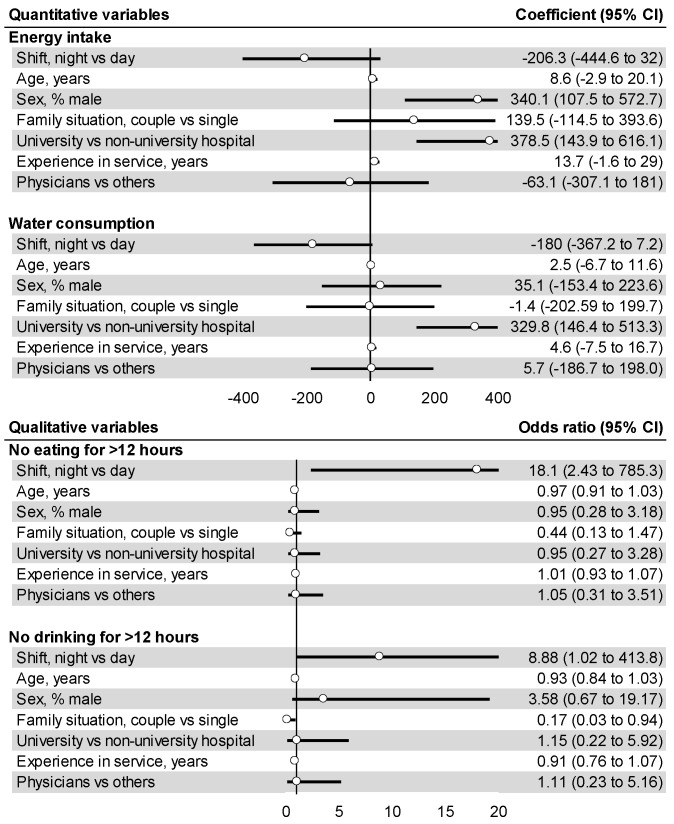
Univariate analysis that found a strong tendency on the impact of night shifts on energy intake, water consumption. Coefficient up to 0 induce a positive impact of the variable (right side of the reference line). Negative coefficient induce a negative impact of the variable (left side of the reference line); 95% CI: 95% confidence interval.

**Table 1 nutrients-14-00829-t001:** Sociodemographic characteristics. SD = standard deviation, n = number of participants, cm = centimeters, kg = kilograms, kg/m^2^ = kilogram per square meter.

	All	Male	Female
	*n* = 184	*n* = 81	*n* = 103
**Age**, years	37.2 ± 10.2	37.8 ± 10.4	36.8 ± 10.1
**Body Mass Index**, kg/m^2^	23.2 ± 3.9	24.3 ± 3.8	22.3 ± 3.7
**Family situation**			
Married/in couple	127 (69%)	57 (71.3%)	70 (68.0%)
Single	56 (30.4%)	23 (28.7%)	33 (32.0%)
Missing data	1 (0.6%)	1 (1.2%)	-
**Children**			
No children	86 (46.7%)	40 (49.4%)	46 (44.7%)
≥1 children	63 (42.3%)	27 (33.3%)	36 (34.8%)
Missing data	34 (18.4%)	14 (17.3%)	21 (20.4%)
**Seniority**, years			
As a provider	10.5 ± 10	10.5 ± 9.6	10.6 ± 10.4
In the hospital	9.9 ± 9.8	9.0 ± 9.0	10.6 ± 10.3
In the emergency department	6.4 ± 10	5.6 ± 6.5	6.9 ± 8.3

**Table 2 nutrients-14-00829-t002:** Impact of night shift on energy intake and water consumption among emergency health care workers. Kcal = kilocalories, g = gram, mg = milligrams, mL = milliliter, *p*-value with * are significant (if <0.05).

	All	Day-Shift	Night-Shift	Comparisons between Shifts
	*n* = 184	*n* = 101	*n* = 83	*p*-Value
**Energy Intake** (kcal)	1523.1 ± 737.0	1606.7 ± 748.2	1400.4 ± 708.3	0.049 *
**Carbohydrates** (g/day)	172.1 ± 83.5	178.8 ± 81.5	163.3 ± 86.0	0.120
**Lipids** (g/day)	69.1 ± 35.1	75.1 ± 35.9	61.13 ± 32.7	0.030 *
Simple fatty acid	29.3 ± 15.4	31.8 ± 16.1	26.0 ± 13.4	0.056
Monounsaturated fatty acid	23.4 ± 14.4	25.6 ± 14.2	20.7 ± 14.2	0.049 *
Polyunsaturated fatty acid	7.0 ± 4.7	7.7 ± 4.8	6.13 ± 4.5	0.050 *
**Protein** (g/day)	74.5 ± 33.7	80.6 ± 30.3	66.4 ± 36.4	<0.001 *
**Cholesterol** (mg/day)	301.3 ± 240.5	318.7 ± 238.9	278.7 ± 243.0	0.161
**Fiber** (g/day)	15.0 ± 7.6	15.8 ± 7.1	13.9 ± 8.2	0.171
**Calcium** (mg/day)	625.5 ± 274.2	665.6 ± 249.3	573.3 ± 297.9	0.049 *
**Water** (mL/day)	1345.8 ± 516.6	1451.4 ± 496.8	1208.3 ± 513.9	0.010 *

## Data Availability

All relevant data are in this manuscript.

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
