# Peer review of "The Negative Impact of Night Shifts on Diet in Emergency Healthcare Workers"

_nutrients, 2022, doi:10.3390/nu14040829_

Round 1

Reviewer 1 Report

Thankyou for the opportunity to review this manuscript on the diet quantity and quality of HCW during dayshifts and nightshifts. This is an interesting field of research and I am always happy to see more research on the eating patterns of shiftworkers, especially our HCW who have such high job demands. This was an enjoyable paper to read and I have several comments below to strengthen this paper and make sure it adds to the field. 

Introduction:

The introduction is well-written and the narrative is clear, however I have a couple of suggestions for points to add to this narrative to strengthen your argument for the study. Firstly, there is a lot of literature on the eating patterns of nurses during the night, I would recommend referencing reviews by Gupta et al and PepÅ‚oÅ„ska et al as well as many qualitative and quantitative papers. Then there needs to be an argument as to what this study adds to this literature that is not already known. Secondly, there could also be an argument in the introduction about how the options for food are limited in a hospital, especially at night, and this is also why we may expect a difference in quality. 

Methods:

In the statistical analysis section the authors state "From personal
experience and observation, there is at least a 20±20% decrease in food intake during night shift compared to day shift" and have used this to calculate sample size. Is there more information that can be given about how this calculation was done? Using only personal experience does not appear like a valid measure of the difference in food intake and there are papers that have quantified this in nurses. 

Results: 

Figure 2, 3 and table 2 are interesting and clear, however would benefit from a notation of the significant p-values (e.g. asterisk). 

Discussion

  • There are sections in the limitations where "(ADD REF)" notes have been left in. 
  • The statement in the conclusion that: "They eat less, less healthily, and even sometimes some emergency HCWs go for long hours without eating" is not fully supported by your results as you don't have evidence that the long hours without eating is bad for the workers. It may also be that they make up for the lack of energy post-shift and so 24h energy is the same with a nightshift or a dayshift (there is evidence to support that). Further, fasting during the nightshift has shown to have benefits for health as eating during the night can be a challenge to the metabolic system, and so perhaps the long hours without eating are good for the workers (again, there are plenty of refs showing this). 

Author Response

Dear Editor,

My coauthors and I welcomed the review of our Manuscript ID nutrients-1590228 entitled "The negative impact of night shift on diet in emergency healthcare workers We have addressed the comments of the reviewers in a revised manuscript and enclose a point-by-point response.

REVIEWER 1:

Thankyou for the opportunity to review this manuscript on the diet quantity and quality of HCW during dayshifts and nightshifts. This is an interesting field of research and I am always happy to see more research on the eating patterns of shiftworkers, especially our HCW who have such high job demands. This was an enjoyable paper to read and I have several comments below to strengthen this paper and make sure it adds to the field.

[REPLY] Thank you for your positive comment.

Introduction:

The introduction is well-written and the narrative is clear, however I have a couple of suggestions for points to add to this narrative to strengthen your argument for the study. Firstly, there is a lot of literature on the eating patterns of nurses during the night, I would recommend referencing reviews by Gupta et al and Pepłońska et al as well as many qualitative and quantitative papers.

[REPLY] Thank you for your comment. We added the following sentences and the suggested references: “Some findings showed that meal timing and meal size has an impact on cognitive per-formance and subjective sleepiness among night shift workers. Some programs proposed to avoid large meals during the beginning of a nightshift and to opt for a small snack to improve performance during the night [15]” (…) [17,18]. Although nurses have been widely studied, no conclusion can be made and the existing epidemiological evidence on the relationship between night shift work of nurses and their dietary habits is inadequate to draw any definite conclusions [19]

Then there needs to be an argument as to what this study adds to this literature that is not already known.

[REPLY] Thank you for your comment. We wrote again the following sentence to increase the specialty of our population, i.e. the Emergency Department workers. “To our knowledge, there is no studies that assessed the influence of night work on the diet among the population of emergency HCWs, nor in relation to their occupational characteristics.”

Secondly, there could also be an argument in the introduction about how the options for food are limited in a hospital, especially at night, and this is also why we may expect a difference in quality.

[REPLY] Thank you for your comment. We added the following sentence: Furthermore, the availability of food is much less important during the night compared to the day in the hospital. It is rarely fresh but often packaged reheated food with a bad presentation. However, food preparation and presentation appears to influence student consumption of school food and adult perception of school meal quality. With the following reference: PMID: 32232865

Methods:

In the statistical analysis section the authors state "From personal experience and observation, there is at least a 20±20% decrease in food intake during night shift compared to day shift" and have used this to calculate sample size. Is there more information that can be given about how this calculation was done? Using only personal experience does not appear like a valid measure of the difference in food intake and there are papers that have quantified this in nurses.

[REPLY] Thank you for your comment. Those data come from a preliminary nonpublished study we performed. We added this sentence in the section.

Results:

Figure 2, 3 and table 2 are interesting and clear, however would benefit from a notation of the significant p-values (e.g. asterisk).

[REPLY] Thank you for your comment. We changed it.

Discussion

There are sections in the limitations where "(ADD REF)" notes have been left in.

[REPLY] Thank you for your comment. References were added.

The statement in the conclusion that: "They eat less, less healthily, and even sometimes some emergency HCWs go for long hours without eating" is not fully supported by your results as you don't have evidence that the long hours without eating is bad for the workers. It may also be that they make up for the lack of energy post-shift and so 24h energy is the same with a nightshift or a dayshift (there is evidence to support that). Further, fasting during the nightshift has shown to have benefits for health as eating during the night can be a challenge to the metabolic system, and so perhaps the long hours without eating are good for the workers (again, there are plenty of refs showing this).

[REPLY] Thank you for your comment. The new conclusion now reads: We showed that emergency healthcare workers working during nightshift have a negative impact on both the amount and quality of nutrients intake. They seem to eat less and less healthily. Furthermore, sometimes some emergency HCWs go for long hours without eating. One fifth of them do not even drink for more than eight consecutive hours of night shifts. Further studies assessing the food intake during rest after nightshift could be relevant. Preventive policies should target those workers.

We also added the following sentences on the limitation section: “Unfortunately, we did not study the food intake the day following the night shift. We previously showed that night shift has a prolonged effect on some outcomes such as biomarkers of stress [23,24]. Further studies assessing the food intake during rest after nightshift could be relevant.”

Reviewer 2 Report

Bouillon-Minois et al conducted a study between night and day shift workers and looked at the differences in the dietary intake. The study in interesting with novel findings. However, I have some concerns that needs to be addressed. 

First, the way the authors presenting groups are giving an impression that it was a same group with repeated measures (which is not). For example, the authors say "the energy intake was decreased in the night shift worker" this should be phrased as "the night shift group reported lower energy intake compared to the day shift group." Please fix this all over the manuscript.

Also, see the attached file for few more suggestions.

Author Response

REVIEWER 2:

Bouillon-Minois et al conducted a study between night and day shift workers and looked at the differences in the dietary intake. The study in interesting with novel findings. However, I have some concerns that needs to be addressed.

First, the way the authors presenting groups are giving an impression that it was a same group with repeated measures (which is not). For example, the authors say "the energy intake was decreased in the night shift worker" this should be phrased as "the night shift group reported lower energy intake compared to the day shift group." Please fix this all over the manuscript.

[REPLY] Thank you for your comment. We changed it in the manuscript. We hope we did not forget somes.

Also, see the attached file for few more suggestions.

[REPLY] Thank you for your comments. We all addressed in the manuscript.

Round 2

Reviewer 1 Report

Thankyou to the authors for addressing my comments. I have nothing further to add.